# The Pitfalls of Regularization in Off-Policy TD Learning

**Gaurav Manek**
Computer Science Department
Carnegie Mellon University
Pittsburgh, PA 15213
gmanek@cs.cmu.edu

**J. Zico Kolter**
Computer Science Department
Carnegie Mellon University
Pittsburgh, PA 15213
zkolter@cs.cmu.edu

## Abstract

Temporal Difference (TD) learning is ubiquitous in reinforcement learning, where it is often combined with off-policy sampling and function approximation. Unfortunately learning with this combination (known as the *deadly triad*), exhibits instability and unbounded error. To account for this, modern RL methods often implicitly (or sometimes explicitly) assume that regularization is sufficient to mitigate the problem in practice; indeed, the standard deadly triad examples from the literature can be "fixed" via proper regularization. In this paper, we introduce a series of new counterexamples to show that the instability and unbounded error of TD methods is *not* solved by regularization. We demonstrate that, in the off-policy setting with linear function approximation, TD methods can fail to learn a non-trivial value function under *any* amount of regularization; we further show that regularization can induce divergence under common conditions; and we show that one of the most promising methods to mitigate this divergence (Emphatic TD algorithms) may also diverge under regularization. We further demonstrate such divergence when using neural networks as function approximators. Thus, we argue that the role of regularization in TD methods needs to be reconsidered, given that it is insufficient to prevent divergence and may itself introduce instability. There needs to be much more care in the practical and theoretical application of regularization to RL methods.

## 1 Introduction

Temporal Difference (TD) learning is a method for learning expected future-discounted quantities from Markov processes, using transition samples to iteratively improve estimates. This is most commonly used to estimate expected future-discounted rewards (the *value function*) in Reinforcement Learning (RL). Advances in RL allow us to use powerful function approximators, and also to use sampling strategies other than naively following the Markov process (MP). When TD, function approximation, and off-policy training are all combined, learned functions exhibit severe instability and divergence, as classically observed by Williams and Baird III [18], Tsitsiklis and Van Roy [15]. This combination is known in the literature as the *deadly triad* [11, pg. 264], and while many contemporary variants of TD are designed to converge despite the instability, the quality of the solution at convergence may be arbitrarily poor.

A common technique to avoid unbounded error is $\ell_2$ *regularization* [14], i.e. penalizing the squared norm of the weights in addition to the TD error. This is generally understood to bound the worst-case error in exchange for biasing the model and potentially increasing the error everywhere else. When used on three common examples of the deadly triad [6, 18, 11, pg.260], regularization appears to

mitigate the worst aspects of the divergence in practice. Consequently, it has become an essential assumption made by many RL algorithms [1, 10, 12, 19, 20, 21, 8] and is seen as routine and innocuous.

We argue that this perspective on regularization in off-policy TD is fundamentally mistaken. While regularization is indeed reasonably well-behaved and innocuous in classic fully-supervised contexts, the use of bootstrapping in TD means that even small amounts of model bias induced by regularization can cause divergence. This is an oft-ignored phenomenon in the literature, and so we introduce a series of new counterexamples (summarized in Table 1) to show how regularization can have counterintuitive and destructive effects in TD. We show that vacuous solutions and training instability are *not* solved by the use of regularization; that applying regularization can sometimes induce divergence and increase worst-case error; and that Emphatic TD algorithms–which are the most promising solution to this divergence–can themselves diverge when regularized. We finally also illustrate misbehaving regularization in the context of neural network value function approximation, demonstrating the general pitfalls of regularization possible in RL algorithms. Regularization needs to be treated cautiously in the context of RL, as it behaves differently than in supervised settings.

Our counterexamples demonstrate these core ideas:

**TD learning off-policy can be unstable and/or have unbounded error even when it converges.**
Following well-established methods we show there is some off-policy distribution under which TD with linear value function approximation diverges *and* learns a model with unbounded error (even if it were able to converge to the TD fixed point). This concisely demonstrates key features of the training error: the error is small when the distribution is close to on-policy, but the error diverges around specific off-policy distributions. The intuition behind this, explained in Section 3, is that the off-policy[1] TD update involves a projection operation that depends on the sampling distribution and can be arbitrarily far away from the true value. This basic fact has already been established by past work [18, 6], but our example is based upon a particular simple three-state MP, drawn in Figure 1a.

**Regularization cannot always mitigate off-policy training error.** We next introduce regularization into our setting, and show how it changes the relationship between training error and off-policy training. As explained in Section 2, we penalize the $\ell_2$-norm of learned (linear) weights with some coefficient $\eta$; as $\eta$ increases, the learned weights approach zero. However, in **Example 1**, we show that there exists an off-policy distribution such that for any $\eta \geq 0 < \infty$, the regularized TD fixed point attains strictly higher approximation error than the zero solution (i.e., the infinitely regularized point). We call such examples *vacuous*. In other words, *vacuous value functions never do better than guessing zero for all states, for any amount of regularization*.

We further analyze this vacuous example in the context of the algorithm in [21]. In this work, the authors assume the use of regularization to derive bounds on the learned error under off-policy sampling. Although these bounds are technically correct in the case of our counterexample, they are very loose, at about 2000 times the threshold of vacuity. This highlights the challenge of formally relying on regularization to bound model error.

**Small amounts of regularization can cause model divergence or large errors.** There is a general implicit assumption in much ML literature that regularlization monotonically shrinks learned weights. This intuition comes from classic fully-supervised machine learning where it typically holds. But because TD bootstraps value estimates (i.e. learns values using its own output), it is possible for small amounts of bias to be arbitrarily magnified. We dub this phenomenon "small-eta error" and illustrate it in **Example 2**. We relate this to the presence of negative eigenvalues in an intermediate step of the solution and show that, in some settings, the error of the TD solution may be relatively small when applied with no regularization but adding regularization causes the model to have worse error than the zero solution.

One common solution to this problem is to lower-bound $\eta$ to guarantee that reguarlization behaves monotonically. However, we further show that such a lower bound may occur after the point of vacuity: a model that is not vacuous becomes vacuous for any regularization parameter above this lower bound. We also show that it is not always possible to select a single $\eta$ *a priori*, with examples of

---

[1]We consider a sampling distribution to be *on-policy* if it follows the stationary distribution of the MP; we do not explicitly consider a separate policy in this paper.

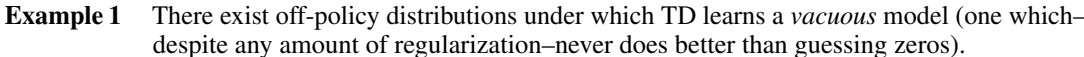

| Example 1 | There exist off-policy distributions under which TD learns a *vacuous* model (one which–despite any amount of regularization–never does better than guessing zeros). |
|---|---|
| Example 2 | Small values of the regularization parameter $\eta$ can make TD diverge in models that otherwise converge. This is an unavoidable effect of bootstrapping in TD, and setting a lower-bound to exclude this may render models vacuous. |
| Example 3 | Emphatic-TD-inspired algorithms are a promising way to reweight samples and mitigate the effects of training off-policy. But if this reweighting is learned using TD, then using regularization can bias the emphasis model and cause the value model itself to diverge. |
| Example 4 | Training instability and increased error due to the deadly triad also occur when neural networks are used. We construct an empirical example and draw qualitative comparisons. |

Table 1: Summary of theorems.

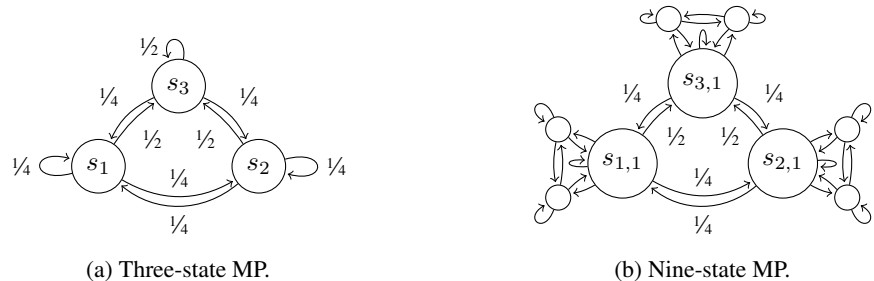

(a) Three-state MP.                      (b) Nine-state MP.

Figure 1: Our three- and nine-state counterexample MPs. We use these to illustrate how TD models can fail despite common mitigating strategies in linear and neural network cases respectively.

mutually-incompatible off-policy distributions where there is no $\eta$ that achieves better than vacuous or nearly-vacuous results at different distributions.

**Emphatic-TD-based algorithms are vulnerable to instability from regularization.** Emphatic-TD [13] fundamentally solves the problem of training off-policy by resampling TD updates so they appear to be on-policy. This technique requires an emphasis model that decides how to scale each TD update, and learning this has been the key challenge preventing widespread adoption of Emphatic-TD. A recent paper [20] proposed learning this emphasis model using "reversed" TD while simultaneously learning the value model using regular TD. The resultant algorithm is called COF-PAC, and employs regularization to ensure that the two TD models eventually converge.

We show that regularization, while necessary, can be harmful for such models in **Example 3**. Specifically, we construct a model that converges to the correct solution without regularization but to an arbitrarily poor solution when regularized. The intuition behind this is that regularizing the emphasis model changes the effective distribution of the TD updates to the value model, which can cause the value model to have arbitrarily large error. We complete the example by showing that regularizing the value function separately does not restore performance.

**Regularization can cause model divergence in neural networks.** So far most analysis of the deadly triad in the literature focuses on the linear case. We extend our example to a nine-state Markov chain (shown in Figure 1b), and show how the previously identified problems persist into the neural network case in **Example 4**. We show two key similarities: first, models trained at certain off-policy distributions may be vacuous. Second, small amounts of regularization counterintuitively *increase* error. This illustrates Example 2 in the NN case.

## 2   Preliminaries and Notation

Consider the $n$-state Markov chain $(\mathcal{S}, P, R, \gamma)$, with state space $\mathcal{S}$, state-dependent reward $R : \mathcal{S} \to \mathbb{R}$, and discount factor $\gamma \in [0, 1]$. $P \in \mathbb{R}^{n \times n}$ is the transition matrix, with $P_{ij}$ encoding the probability of moving from state $i$ to $j$. We wish to estimate the value function $V : \mathcal{S} \to R$, defined

as the expected discounted future reward of being in each state: $V(s) \doteq \mathbf{E}\left[\sum_{t=0}^{\infty} \gamma^t R(s_t) \mid s_0 = s\right]$. A key property is that it follows the Bellman equation:

$$V = R + \gamma P V \tag{1}$$

Using linear function approximation to learn $V$, we assume a matrix of feature-vectors $\Phi \in \mathbb{R}^{n \times k}$ that is fixed, and a vector of parameters $w \in \mathbb{R}^k$ that is learned. The Bellman equation is then:

$$\Phi w = R + \gamma P \Phi w \tag{2}$$

When $w$ is learned with TD, this equation is only valid if the TD updates are *on-policy* (that is, they are distributed according to the steady-state probability of visiting each state, written as $\pi \in \mathbb{R}^n$). In the general case, where TD updates follow a (possibly) different distribution $\mu \in \mathbb{R}_0^n$, the TD solution is a fixed point of the Bellman operator followed by a projection [6]:

$$\Phi w = \Pi_\mu \left( R + \gamma P \Phi w \right) \tag{3}$$

where the matrix $\Pi_\mu = \Phi(\Phi^\top D \Phi)^{-1} \Phi^\top D$ projects the Bellman backup onto the columnspace of $\Phi$, reweighted by the diagonal matrix $D = \text{diag}(\mu)$. This yields the closed-form solution:

$$w = A^{-1} \vec{b} \tag{4}$$

Where $A = \Phi^\top D (I - \gamma P) \Phi$ and $\vec{b} = \Phi^\top D R$. When this solution is subject to $\ell_2$ regularization, some non-negative $\eta$ is added to ensure the matrix being inverted is positive definite:

$$w^*(\eta) = (A + \eta I)^{-1} \vec{b} \tag{5}$$

As will be important later, we note that as $\eta$ increases it drives $w^*(\eta)$ towards zero.

## 3 Our Counterexamples

When deadly triad conditions are present, TD may learn a value function with arbitrarily large error even if the true value function can be represented with low error. Consider the three-state MP in Figure 1a, which we instantiate with the value function $V = [1, \, 2.2, \, 1.05]^\top$ and discount factor $\gamma = 0.99$. The reward function is computed as $R \leftarrow (I - \gamma P)V$. We choose a basis $\Phi$ with small representation error $\|\Pi_\mu V - V\| \leq \epsilon$:

$$\Phi = \begin{bmatrix} 1 & 0 \\ 0 & -2.2 \\ 1/2(1.05 + \epsilon) & -1/2(1.05 + \epsilon) \end{bmatrix} \qquad \text{where } \epsilon > 0 \tag{6}$$

We first consider the unregularized ($\eta = 0$) case, closely following the derivation in [6]. We wish to show there is some sampling distribution $\mu$ such that error in the learned value function is unbounded. To do this, we set $\mu = [0.56(1-p), 0.56p, 0.44]$, where $p \in (0, 1)$. We set $\epsilon = 10^{-4}$ and find $p$ around which $A$ is ill-conditioned by solving $\det(A) = 0$:

$$p = 0.102631 \qquad \vee \qquad p = 0.807255 \tag{7}$$

$A^{-1}$ (and consequently the error) can be made arbitrarily large by selecting $p$ close to these values, which completes the introductory example. Now we look at the behavior of TD under regularization, which is the main contribution of our paper.

### 3.1 Regularization cannot always mitigate off-policy training error.

There is a belief in the literature that regularization is a trade-off between reducing the blow-up of asymptotic errors and accurately learning the value function everywhere else [1, 21]. However, this belief does not accurately capture the nature of regularization: we show that it is possible to learn models that never perform better than always guessing zero despite any amount of regularization. That is, the TD error at all $\eta$ is at least as much as the error as $\eta \to \infty$. We call such models *vacuous*.

**Example 1.** When TD is regularized, there may exist some off-policy distribution at which TD learns a vacuous model. In notation:

$$\|\Phi w^*(\eta) - V\| \geq \lim_{\eta \to \infty} \|\Phi w^*(\eta) - V\| = \|\Phi \vec{0} - V\| = \|V\| \qquad \forall \eta \in \mathbb{R}_0^+ \qquad (8)$$

*Details.* We use the same setting as in Section 3. A detailed derivation is provided in Appendix B.2.

We begin by noting that we can easily find the solution $\hat{w}$ that minimizes the least-squares error $\|\Phi \hat{w} - V\|$. If we consider this solution as a vector (as drawn in Figure 2a), we can immediately see that there is an $\ell_2$-ball around $\hat{w}$ corresponding to the set of $w^*(\eta)$ with no more than $\|V\|$ error.

Similarly, we can trace the trajectory that the TD solution $w^*(\eta)$ takes as $\eta$ is increased from 0 to $\infty$. We know that, as $\eta \to \infty$, $w^*(\eta)$ is crushed to zero and so all trajectories must eventually terminate at the origin. When regularized models are not vacuous, the trajectory intersects the non-vacuous-error ball. We see this in trajectory 2, where the error briefly dips below $\|V\|$ in Figure 2b.

Intuitively, a sufficient condition for a solution to be vacuous is that it remains in the half-space that is tangent to and excludes the non-vacuous parameter ball. This is equivalent to finding some distribution $\mu$ such that $\hat{w}^\top w^*(\eta) \leq 0$ for all $\eta$, which we numerically solve to obtain the model in trajectory 1. From Figure 2a we can see the trajectory remains in the half-space, and from Figure 2b we can see that the error is never less than $\|V\|$. Trajectory 1 is a vacuous example. □

We observe that Example 1, because it remains entirely in the halfspace $\hat{w}^\top w^*(\eta) \leq 0$, could easily be generalized to other forms of regularization. We leave this for future work.

***Breaking the Deadly Triad* and our counterexample.** In light of our example we examine the work of [21] in which the authors derive a bound for the regularized TD error under a novel double-projection update rule. We apply our example to their bound and show that their method may produce loose bounds on TD solutions, and so doesn't quite break the deadly triad:

$$\|\Phi w^*(\eta) - V\| \leq \frac{1}{\xi} \left( \frac{\sigma_{\max}(\Phi)^2}{\sigma_{\min}(\Phi)^4 \sigma_{\min}(D)^{2.5}} \cdot \|V\|\eta + \|\Pi_D V - V\| \right) \qquad (9)$$

for $\xi \in [0,1]$, where $\sigma_{\max}$ and $\sigma_{\min}$ denote the largest and smallest singular value respectively. Theorem 2 from [21] bounds $\eta$, and therefore also $b$:

$$\eta > \arg \inf_{\eta} \|\Phi - C_0\| = {}^{0.177}/(1-\xi)^2 \qquad (10)$$

$$\inf_{\xi} b(\xi, \eta) = 5.20 \times 10^4 \approx 2000 * \|V\| \qquad (11)$$

Their method bounds the error in our example by $2000 * \|V\|$, which is tremendously loose. (We analyze a different example in Appendix B.3, showing a still-loose but improved bound of $8 * \|V\|$.) This illustrates the risk of relying on regularization to formally bound model error.

## 3.2 Small amounts of regularization can cause large increases in training error.

There is a general assumption in the literature that $\ell_2$ regularization monotonically shrinks the learned weights. While this is true in classification, regression, and other non-bootstrapping contexts, this is not true in TD. Because TD bootstraps values, it is possible for model bias to be arbitrarily magnified.

This can be understood in terms of the eigenvalues of the matrix $A$ in Equation 5. By increasing values along the diagonal, $\ell_2$ regularization increases eigenvalues of the matrix $(A + \eta I)$ to ensure it is positive definite. Under off-policy distributions, it is possible for $A$ to have eigenvalues that are negative or zero. This implies that there are $\eta$ for which $\det(A + \eta I) = 0$, and selecting $\eta$ close to these values allows us to achieve arbitrarily high error. We show one such case in Example 2. This is not merely theoretical–we demonstrate this in the neural network case in Section 3.4.

**Example 2.** When TD is regularized, the model may diverge around (typically small) values of $\eta$. Lower-bounding $\eta$, a common mitigation, can make well-behaved models vacuous. It is not always possible to select a single value of $\eta$ that makes models vacuous at different sampling distributions.

*Details.* Using our three-state example, we set $\mu = [0.05, 0.05, 0.9]$ and solve for $\det(A + \eta I) = 0$:

$$0 = \det(A + \eta I) = \eta^2 + 5.45 \times 10^{-2}\eta - 7.47 \times 10^{-3} \implies \eta = 0.0634 \qquad (12)$$

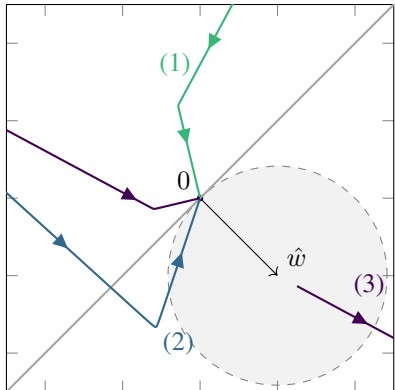 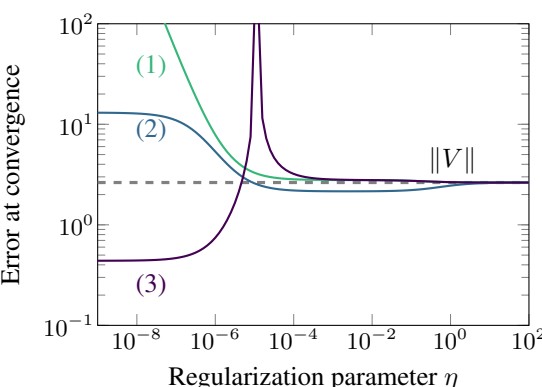

(a) As $\eta$ increases, $w^*(\eta)$ traces different trajectories at different $\mu$. $\hat{w}$ minimizes the error, and we shade the area with TD error less than $\|V\|$.

(b) We plot the error curves corresponding to the three $w^*(\eta)$ trajectories, along with $\|V\|$. Trajectory 1 is vacuous because the error is at least $\|V\|$ for all $\eta$.

Figure 2: Plotting the trajectory of the parameters on the left and the errors on the right, we show how our counterexample 1 is never better than $\|V\|$ because it remains in halfspace where $\hat{w}^\top w^*(\eta) \leq 0$. For comparison, we show trajectory 2 that is improved by regularization, and 3, which exhibits small-$\eta$ errors. (The trajectories are stretched, so the errors in the two plots are not directly comparable.)

As in the introductory example, the error can be made arbitrarily large by setting $\eta \approx 0.0634$.

This small-$\eta$ divergence effect can appear in several ways, illustrated in Figure 3a. Typically, this appears as one or more points at which TD error diverges before the region at which regularization reduces the model error below $\|V\|$. The first and second plot in Figure 3a show two such cases, where the error increases sharply at two and one points respectively.

In the literature, it is commonly assumed that $A$ is "nearly" positive definite, where only a few eigenvalues are non-positive, and those are close to zero. This gives rise to the common mitigation of setting a lower-bound $\eta_0$ such that $(A + \eta I)$ is positive definite for $\eta > \eta_0$. This may render an otherwise well-behaved model vacuous. The third plot in Figure 3a illustrates this: the model is not vacuous when unregularized, but is vacuous in the domain $\eta > 10^{-2}$ where divergence is prohibited.

A common practice in the literature is to set $\eta$ before training, without regard for the sampling distribution. This is ill advised, as the value may be under- or over-regularizing depending on the sampling distribution. One such example is illustrated in Figure 3b, where selecting an $\eta$ that minimizes the error for one distribution will lead to vacuous or nearly-vacuous results in the other two. A second example in Figure 2b has no single $\eta$ for which trajectories 2 and 3 are both non-vacuous. This is especially relevant as regularization is commonly used to permit distribution drift during training, as discussed in Section 4. If the training distribution changes while $\eta$ is fixed, then algorithms that can be proven to converge to good solutions under some original distribution may converge to poor solutions as the distribution drifts. □

### 3.3 Emphatic approaches and our counterexample

Emphatic-TD eliminates instability from off-policy sampling by reweighting incoming data (via an importance function) so it appears to be on-policy. There is considerable interest in making this more practical, especially by learning the importance and value models simultaneously. A leading example of this work is COF-PAC [20], which uses $\ell_2$-regularized versions of GTD2 [12] to learn both the value and emphasis models. The authors rely on regularization, particularly because the target policy changes during learning. This makes COF-PAC vulnerable to regularization-caused error. We illustrate this with Example 3 in which COF-PAC learns correctly when unregularized, but has large error when regularized. (Mathematical details are deferred to Appendix B.5.)

**Example 3.** COF-PAC may learn the value function with low error when unregularized, but with arbitrarily high error when regularized.
*Details.* Conceptually, COF-PAC maintains two separate models that are each updated by TD: the emphasis and the value models. This emphasis model is used to reweight TD updates to the value

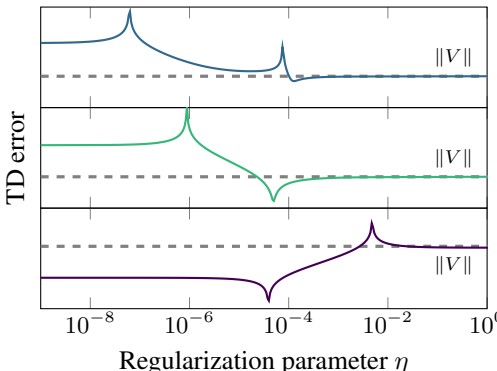
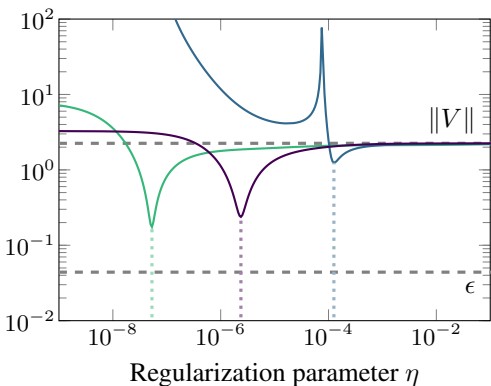

(a) Different MPs at off-policy distributions selected to show small-$\eta$ error. The error may increase at multiple $\eta$, and may even occur *after* the optimal $\eta$.

(b) Three off-policy distributions with mutually incompatible $\eta$. There is no $\eta$ at which all models are not vacuous or nearly vacuous.

Figure 3: We plot TD error against $\eta$ to show small-$\eta$ errors (left) and mututally-incompatible $\eta$ (right). We also plot the error at the limit of vacuity $\|V\|$ and the representation error $\epsilon$.

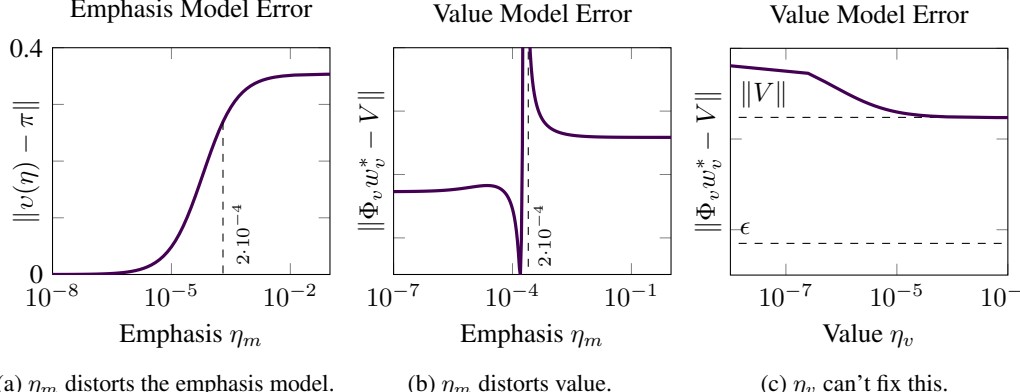

(a) $\eta_m$ distorts the emphasis model.

(b) $\eta_m$ distorts value.

(c) $\eta_v$ can't fix this.

Figure 4: Regularization on the emphasis model ($\eta_m$) distorts the effective distribution (Figure 4a). Specific values of $\eta_m$ induce the value function to diverge (Figure 4b). The resultant value function is vacuous (Figure 4c). Under COF-PAC, regularization can greatly increase model error.

function so they appear to come from the on-policy distribution. Our strategy is to first show how regularization biases the emphasis model, and then how this bias causes the value model to diverge. We begin with our three-state MP, noting its on-policy distribution is $\pi = [.25\ .25,\ .5]$. We wish to learn the values using COF-PAC while sampling off-policy at $\mu = [.2\ .2\ .6]$.

Now we introduce a key conceptual tool: $\upsilon(\eta_m)$, which is the effective distribution seen by the TD-updates, influenced by the emphasis regularization parameter $\eta_m$. Unregularized, COF-PAC is able to resample off-policy updates to the on-policy distribution: $\upsilon(0) \equiv \pi$. If the model is regularized, then the effective distribution moves away from $\pi$. Figure 4a illustrates the distance between $\upsilon(\eta_m)$ and $\pi$ as the regularlization parameter increases.

We can use the effective distribution to compute the error in the value model. Plotting the relationship between the value function error and $\eta_m$ in Figure 4b, we can see the value function has asymptotic error around $\eta_m = 2 \times 10^{-4}$. This shows how COF-PAC may diverge with specific regularization.

COF-PAC also allows for the value function to be separately regularized with parameter $\eta_v$. We show the effect of this in Figure 4c, where the value function never does much better than $\|V\|$ making it (nearly) vacuous. We can conclude that regularizing the emphasis model may cause the value model to diverge, and this cannot be fixed by regularizing the value function separately. $\qquad\square$

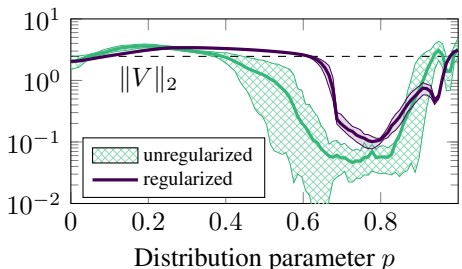
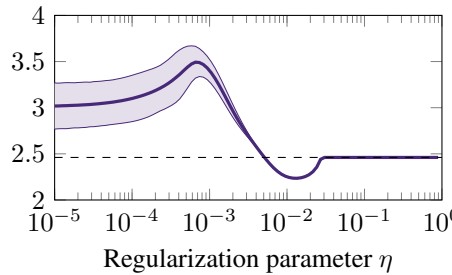

(a) Mean and $10^{\text{th}}$–$90^{\text{th}}$ percentile errors of 100 NN value models trained to convergence.

(b) The relationship between error and $\eta$ at the off-policy distribution $p = 0.31$.

Figure 5: We illustrate how regularization interacts with NN value functions, showing that the problems identified in this paper persist in the NN case.

COF-PAC makes the strong assumption that Kolter's relaxed-contraction condition [6, eqn. 10] holds in both the emphasis and value models [20, asm. 4]. We discuss this in Appendix B.5.1.

### 3.4 Applied to multi-layer networks

We use a 9-state variant of our example to study the deadly triad in multi-layer neural networks (NNs). A deterministic observation function is chosen so we can control the degree of function approximation. We train a simple two-layer neural network with 3 neurons in the hidden layer. The value function is assigned pseudo-randomly in range $[-1, 1]$. (See Appendix C for details.)

**Example 4.** Vacuous models and small-$\eta$ error also occur in neural network conditions.

*Details.* We train 100 models using simple semi-gradient TD updates under a fixed learning rate. We plot the mean and the $10^{\text{th}}$–$90^{\text{th}}$ percentile range in Figure 5a, with and without regularization. TD is known to exhibit high variance, and regularization is the traditional remedy for that. We corroborate this by noting that the performance of the unregularized model varies widely, but regularization leads to similar performance across initializations at the cost of increased error.

First, we show that vacuous models may exist in the neural network case. In Figure 5a, note how there are some off-policy distributions under which both the regularized and unregularized models perform worse than the threshold of vacuity. We can numerically verify that vacuous models exist. Second, we show the small-$\eta$ error problem in the neural network case in Figure 5b, where we plot the TD error against $\eta$ at a fixed off-policy distribution. We observe that around $\eta \approx 10^{-3}$ the TD Error unexpectedly *increases* before decreasing, which clearly illustrates this phenomenon. $\square$

These qualitative links show a clear connection between the neural network case and the linear case, and highlights the importance of correctly handling off-policy sampling.

## 4   Related Work

Three examples of the deadly triad are common in the literature: the classic Tsitsiklis and Van Roy $(w, 2w)$ example [11, p. 260], Kolter's example [6], and Baird's counterexample which shows how training instability can exist despite overparameterization [18].

$\ell_2$ regularization is common when proving that an algorithm converges under a changing sampling policy. This is seen in GTD (analyzed in [19]), GTD2 [12], RO-TD [10], and COF-PAC [20]. This assumption may also be used to ensure convergence when training with a target network [21]. Despite the prevalence of regularization, the induced bias from using it is not well studied. It is often dismissed as a mere technical assumption, as in [1]. We contradict that: using regularization for convergence proofs may induce catastrophic bias. By showing concrete examples, this work hopes to inspire further investigation into regularization-induced bias in the same vein as [19].

**Alternatives to regularization and TD**   We focus on $\ell_2$ regularization in this paper, which penalizes the $\ell_2$-norm of the learned weights; it is also possible to use $\ell_1$ regularization with a proximal

operator/saddle point formulation as in [10], or any convex regularization term under a fixed target policy [19]. Instead of directly regularizing the weights, COP-TD uses a discounted update [4]. DisCor [7] propagates bounds on Q-value estimates to quickly converge TD learning in the face of large bootstrapping error; it is not clear if DisCor can overcome off-policy sampling. A separate primal-dual saddle point method has also been adapted to $\ell_2$ regularization [2] and is known to converge under deadly triad conditions, and recent work [17] has derived error bounds with improved scaling properties in the linear setting, offering a promising line of research.

Emphatic-TD [13] fixes the fundamental problem in off-policy TD by reweighting updates so they appear on-policy. The core idea underlying these techniques is to estimate the "followon trace" for each state, the (weighted, $\lambda$- and $\gamma-$discounted) probability mass of all states whose value estimates it influences. This trace is then used to estimate the emphasis, which is the reweighting factor for each update. While this family of methods is provably optimal in expectation, it is subject to tremendous variance in theory and practice, especially when the importance is estimated using Monte-Carlo sampling.[2] In practice, these methods learn the follow-on trace using TD [5, 20] or similar [16], which makes them vulnerable to bias induced by the use of regularization.

## 5   Conclusion

There is a tremendous focus in the RL literature on proving convergence of novel algorithms, but not on the error at convergence. Papers like [21] are laudable because they provide error bounds; even if the current bounds are loose, future work will no doubt tighten them. In this work, we show that the popular technique of $\ell_2$ regularization does not always prevent singularities and could even introduce catastrophic divergence. We show this with a new counterexample that elegantly illustrates the problems with learning off-policy and how it persists into the NN case.

Even though regularization can catastrophically fail in the ways we illustrate, it remains a reasonable method that may offer a fair tradeoff – as long as we are careful to check that we are not running afoul of the failure modes we explain in the paper. It may be possible to design an adaptive regularization scheme that can avoid these pathologies. For now, testing the model performance over a range of regularization parameters (spanning several orders of magnitude) is the best option we have to detect such pathological behavior.

Emphatic-TD is perhaps the most promising area of research for mitigating off-policy TD-learning. The key problem preventing its widespread adoption is the difficulty in estimating the emphasis function, but future work in this area may be able to overcome this. Our example shows the risk of relying on regularization in practical implementations of such methods. It is absolutely critical that Emphatic algorithms correctly manage regularization to avoid the risks that we highlight in this paper.

---

[2]Sutton and Barto's textbook [11] says about Emphatic-TD that "it is nigh impossible to get consistent results in computational experiments." (when applied to Baird's example).

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
