# OpenReview forum: "The Pitfalls of Regularization in Off-Policy TD Learning"
_NeurIPS.cc/2022/Conference — NeurIPS 2022 Accept_

### Official Review · Reviewer_cP3d · 2022-07-07

**Rating:** 7
**Confidence:** 3
**Soundness:** 3 good
**Presentation:** 4 excellent
**Contribution:** 3 good

**Summary:**

Off-policy RL can be divergent when function approximation and bootstrapping are utilized, and it is commonly believed that $l^2$ regularization prevents divergence in this case. The authors show that there are problems where linear models with any amount of $l^2$ regularization produce worse error than the trivial zero solution does. An analytical and geometric argument is presented, and experiments on linear and nonlinear models demonstrate specific examples in which $l^2$ regularization can increase value-estimation error or lead to divergence.

**Questions:**

1. In Example 1, how can you guarantee that $\hat{w}^Tw^*(\eta) \leq 0$, $\forall~\eta$? Do you have any theoretical results for when a distribution $\mu$ that causes this either exists or does not exist?
1. The geometric interpretation of a vacuous model (a trajectory within the halfplane tangent to the $l^2$ ball) makes sense for linear models, but I don’t see why it would necessarily extend to nonlinear (NN) models. Could you elaborate on why you think these results are transferrable?
1. In Figure 5b, what would happen if you used a NN with more than just 3 neurons in each layer? Would you still expect to see an increase in error for some $\eta$-values?


**Strengths And Weaknesses:**

**Strengths**
- The paper addresses a fundamental issue in off-policy RL with function approximation, with the potential for significant practical implications. I expect these findings will be of interest to a wide audience of RL researchers.
- The paper is very well written. There are lots of motivating examples presented in a logical, cohesive flow---almost like reading a chapter of a textbook.
- The geometric interpretation in Figure 2a was helpful for developing an intuition for the problem addressed by the authors.

**Weaknesses**
- Overall, I finished reading the paper feeling a lack of clear instruction about “the pitfalls of regularization.” The authors’ general takeaway seems to be that regularization is bad and should be avoided. In reality, the issue seems to be far more nuanced than that. The paper would benefit from further discussion about when regularization is good, too, and what the practical trade-offs may be. For instance, it was not clear to me how pertinent the specific counterexamples in the paper are to off-policy RL in general.
- I am not totally convinced by the neural network (NN) experiment in Example 4, Figure 5b. The authors claim that their divergence results in the linear approximation case carry over to nonlinear approximation as well, since there are some $\eta$-values that increase the error of the learned NN value function. In reality, I think this conclusion is based somewhat on the misleading x-axis log scale, and the fact that the y-axis does not start at the origin. To me, it looks like, below $\eta=10^{-3}$, there is effectively no regularization, and the error is a plateau. If the x-axis were linear, then this region would be essentially negligible. Then, at $\eta=10^{-3}$, after a short uptick in error, the error starts to decrease rapidly for about two orders of magnitude, before the regularization becomes too strong and starts to worsen again. The takeaway I ultimately got from this graph is that a moderate amount of regularization is actually quite helpful!

**Minor Edits**
- Line 48: The grammar in this sentence sounds incorrect to me.
- Line 96: Typo in “regularization.”
- Line 129: Don’t end a paper section with a colon.
- Lines 145, 149: I would change the notation “trajectory (1),” since it looks like you are referencing an equation there.
- Line 195: I think it should be “reweighting,” not “reweighing.”

---

> ### Author Response · Authors · 2022-08-01
> **Response**
>
> Thank you for the detailed review! We appreciate the attention and effort that went in to it.
>
> To answer your questions:
>
> 1. We don't have any broader theoretical results on more generic classes of scenario where these pathologies exists. The best intuition we can offer is that it tends to happen when the $A$-matrix has at least one negative eigenvalue and $\mu$ is far from the on-policy distribution.
>
> 2. The geometric intuition unfortunately doesn't extend cleanly to the NN case -- the relationship between the parameters and the output is nonlinear, and so the "non-vacuous" region no longer has a nice, convex shape that provides an intuitive explanation.
>
> 3. We ran a few more experiments on two-layer networks with a range of parameters covering both under- and over-parameterization regimes. As expected, when under-regularized, having more parameters reduces error. We also show that the behavior around the small-$\eta$ divergence does not change, even when the hidden layer is stretched to 64 parameters. (For comparison, the MDP only has 9 states.) We include this in the new Appendix C.1, with plots in Figure 10.
>
> In addition to your questions, you raise an important point about regularization: its not all doom-and-gloom, and it does have an important part in training RL algorithms. Our paper was intended to point out that regularization behaves differently (and counterintuitively!) with bootstraping than in fully-supervised contexts, and to emphasize that regularization should be treated with caution. We hope to address a pattern in RL where regularization is used without special attention to the possible errors and failure modes it may introduce. We've added this to the discussion and conclusion, and also mention that trying different regularization parameters, adaptive regularization schemes, or even just changing regularization schedules can all be used to mitigate some of the problems we identify.
>
> In Figure 5b, the increase in error is strong evidence of small-eta divergence in the NN case. To make this more apparent, we evaluated the same MDP at a different off-policy distribution, which is included in the updated appendix as Figure 9. At this other off-policy distribution, the optimal $\eta$ is about $10^{-3}$, and divergence occurs _after_ that, which should address the concern that $\eta \approx 10^{-3}$ is negligibly small. Interestingly, these two examples are mutually incompatible (that is, there is no single $\eta$ for which both examples are simultaneously not vacuous), which further emphasizes the need for either adaptive regularization or even just trying different parameters for each trained model.
>
> To answer the more general question about how much this applies to general off-policy RL, we don't know and are also curious about this. The offline/batch RL literature contains many examples of how on-policy RL algorithms diverge in the face of off-policy sampling, so the algorithms are vulnerable. There is some evidence to suggest that RL algorithms that maintain transition buffers (e.g. SAC/MBPO) yield better performance if out-of-distribution "stale" transitions are evicted from the buffer. We added this discussion to the new Appendix C.2, and are considering including it in Section 3.4.
>
> (We'll also fix the minor typos/grammatical errors you flagged. Thank you!)

---

> > ### Comment · Reviewer_cP3d · 2022-08-07
> > **Re: Response**
> >
> > I thank the authors for their additional experiments and other updates to the paper. The newly added discussions help to acknowledge and contextualize the limitations of the findings within modern RL (neural networks, adaptive regularization techniques, etc) which I think makes the message of the paper stronger and more useful to readers overall. My score remains the same, and I continue to recommend acceptance of this interesting work.

---

### Official Review · Reviewer_zUeM · 2022-07-10

**Rating:** 6
**Confidence:** 3
**Soundness:** 3 good
**Presentation:** 3 good
**Contribution:** 2 fair

**Summary:**

This paper proposes and analyzes a set of simple counter examples to show that l2 regularization can not mitigate the problems faced value estimation in the deadly triad setting consisting of off-policy learning under function approximation with bootstrap. The authors also analyze emphatic TD approaches and show that l2 regularization on the weight norm is insufficient.



**Questions:**

* What is the precise definition of a "vacuous model"? Anything that achieves an error worse than the trivial zero-prediction solution? An alternate threshold for vacuity could be the variance of the value function as opposed to the l2 norm. Can the authors make this a bit more clear as this term is repeatedly used through the paper across multiple sections?

* The takeaways from Section 3.4 are a bit unclear. It appears to mainly imply that there are situations where the resulting solution from TD is not useful. What could be more interesting is some evidence that these phenomena can occur with neural nets occur outside of the specific counter examples considered.

* What is the distribution parameter p in Fig 5(a)?

* Nit: small-eta in L230.

**Limitations:**

Adequate

**Strengths And Weaknesses:**

Strengths:

* This is a well written paper with simple but non-trivial examples that demonstrate concretely that l2 regularization can not solve the convergence issues highlighted in the "deadly triad" problem in off-policy learning under function approximation.

* The analysis over a range of regularization strengths and comparison against "vacuous bounds" helps understand when prior bounds like Eq (9) are not informative.

* The analysis showing how the off-policy distributions can influence the optimum regularization, specifically causing different non-overlapping ranges of $\eta$ to be necessary for non-vacuous performance seems like a helpful qualitative insight even though it arises out of very specific counter examples.

Weaknesses:

* I'm not sure that there was a widespread belief that regularization necessarily solves the deadly triad, so in that sense the demonstrated phenomenon is not necessarily surprising even if valuable

* The paper is presented as a general study of regularization though the results are specific to l2 weight regularization.

---

> ### Author Response · Authors · 2022-08-01
> **Response**
>
> Thank you for the time and work in writing your review!
>
> We agree that people don't always necessarily believe that regularization solves the deadly triad. Instead, our paper comes from the observation that a common pattern in this area is to use regularization without special attention to the possible errors it may introduce. In that sense, our paper points out that regularization behaves differently (and counterintuitively!) with bootstraping than in fully-supervised contexts, emphasizes that regularization is not a costless decision, and encourages caution in using it.
>
> Our paper focuses on L2 regularization because it is by far the most widely used, and the kinds of errors we identify (vacuous models, small-eta divergence, etc.) all analogously apply to L1 regularization.
>
> To answer your questions: A vacuous model is one that does no better than the trivial zero solution regardless of the amount of regularization. We added extra text in the introduction to make this clear. The key takeaway from Section 3.4 is that the problems we've identified so far are not mere artifacts of linear approximation or caused by a unique pathological basis. Instead, these also apply when the basis is not fixed (i.e. the NN case).
>
> The more general question is if these failure modes apply to modern RL algorithms on benchmark and real-world tasks. We don't know if that happens, and are also very curious about that. The various ingredients are in place: we know (thanks to the offline/batch RL literature) that the algorithms are vulnerable to this on static datasets, and there is some evidence to suggest that RL algorithms that maintain transition buffers (e.g. SAC/MBPO) suffer if the transition buffer is too large (suggesting that the presence of out-of-distribution "stale" transitions decreases performance). However, its not clear to us how we can prove this is the failure mode. (We included this in the new Appendix C.2 and may later move it to Section 3.4.)
>
> ($p$ in Fig. 5 is described by Eqn. 40 in the appendix, we'll clarify that and fix L230. Thank you!)

---

> > ### Comment · Reviewer_zUeM · 2022-08-08
> > **Response**
> >
> > Thank you for the response. I have edited my review/score upward after reading your clarifications.

---

### Official Review · Reviewer_NoAj · 2022-07-11

**Rating:** 7
**Confidence:** 4
**Soundness:** 4 excellent
**Presentation:** 4 excellent
**Contribution:** 4 excellent

**Summary:**

It is well-known that TD methods diverge when used in
conjunction with value function approximation and off-policy learning.
Several recent works have proposed the use of additional regularization to
prevent divergence in this setting. This paper provides several counterexamples
demonstrating that adding regularization to linear TD methods can lead to
vacuous value estimates, and that regularization can in fact cause divergence
in certain settings. Counterexamples are also given for a popular emphatic TD algorithm,
and for non-linear TD. Taken together, the results suggest that the role of regularization
being proposed by recent works needs reconsideration.


**Questions:**

- Emphatic approaches:
  - The conclusions in section 3.3 hold only for a specific
    algorithm, COF-PAC, which simultaneously learns both a
    value model and emphasis model, but not necessarily to
    emphatic methods in general; am I understanding this correctly?
  - Do you expect similar conclusions could hold under more general assumptions
    on the emphasis model? or is the core problem specifically the *learning* of the emphasis model
    in parallel with the value model?


**Limitations:**

yes

**Strengths And Weaknesses:**

Overall, I thought the paper was excellent --- easily the best I reviewed this round of reviews --- and recommend acceptance.
The results are novel, thorough, and clearly relevant to
on-going discourse in the community; the
paper is well-written and the explanations were clear.
I really only have minor comments and questions, which I provide below.

### Minor Comments
- Line 266: "A separate primal-dual saddle point method has also been adapted to $\ell_2$ regularization,
  but error bounds at convergence are not yet known"
  - I think this may no longer be true; the results in Du et al. (2022) capture
    the regularized MSPBE as a special case. This paper was posted after the NeurIPS deadline I believe,
    but I think it is worth revising this sentence. I don't think the existence of such bounds for
    the $\ell_2$ regularized MSPBE takes anything away from the results presented here anyways
- Line 158, typo: $\xi\in[0..1]$ should read $\xi\in[0,1]$

---

> ### Author Response · Authors · 2022-08-01
> **Response**
>
> We appreciate the time and effort that went into your review!
>
> Thank you for pointing out Du et al. 2022 -- we'll read through the paper and update L266 to reflect this new information.
>
> The conclusions in our Section 3.4 should apply to any Emphatic-TD-based algorithm that learns the emphasis function (1) via TD/bootstrapping, and (2) uses regularization. COF-PAC was chosen because it is an exemplar of these: it uses "time-reversed" TD to estimate the emphasis, and it (like almost every other work that offers performance bounds) uses regularization to ensure that the models converge despite slowly changing sampling distributions.
>
> (We will also correct L158, thanks!)

---

> > ### Comment · Reviewer_NoAj · 2022-08-03
> > **reference**
> >
> > Sorry, I just realized I forgot to actually include the reference I was talking about. It sounds like you found it already, but here's the paper I was talking about just in case:
> >
> > Du, S. S., Gidel, G., Jordan, M. I., & Li, C. J. (2022). Optimal Extragradient-Based Bilinearly-Coupled Saddle-Point Optimization.

---

### Official Review · Reviewer_EBcQ · 2022-07-14

**Rating:** 7
**Confidence:** 3
**Soundness:** 3 good
**Presentation:** 3 good
**Contribution:** 2 fair

**Summary:**

This submission is a theoretical paper which investigates a common belief in the community around the problem of the deadly triad: regularization would prevent instability and divergence of TD methods with function approximation in an off-policy setting. This paper provides four counter-examples showing that this is not the case and that l2 regularization can lead to instability and divergence in the context of function approximation and even with neural networks. The paper also shows that emphatic methods can also suffer from these divergence issues. This paper is an invitation to reconsider the use of regularization in off-policy TD learning.


**Questions:**

What are promising solutions to address the issues highlighted in the paper? While regularization can be harmful as the paper shows, itsnt there a tradeoff given it can help in other examples under the same off-policy setting?

**Limitations:**

There is no dedicated section to limitations. I would say this work highlights issues in the off-policy setting but is limited in terms of practical solutions.


Post-rebuttal update:

I want to thank the authors for engaging into this discussion. After carefully reading other reviews, I will update my score to 7 and recommend acceptance.

**Strengths And Weaknesses:**

**Contribution**
The paper provides interesting new counter-examples showing some issues of TD methods under regularization. I also appreciated the explanatory figures 2, 3, 4 , 5 illustrating the counter examples from the text.
As mentioned by the submission, the fact that off-policy TD learning can be unstable or have unbounded error when it converges is already know in the literature but the authors provide a simpler example. The other insights and examples from sections 3.2, 3.3 and 3.4 are new as far as I am aware.

It would be interesting to understand whether these counter examples could have an impact on current RL algorithms that have been used on common benchmarks, ie are these counter examples extreme situations or can they explain empirical results from past papers?

In particular, the analysis from section 3.4 is limited to two-layer neural networks and it could be interesting to see if the insights would hold for overparametrized architectures too. This shouldn't prevent acceptance of the paper though.
One weakness I see is that given the dynamics of sgd with off-policy TD are unknown it is hard to conclude that the issues highlighted by the paper are a practical issue for current deep rl algorithms. This aspect is briefly mentioned by the authors line 246.

**Organization and Clarity**
The paper is overall well-written and well-organized.
L70 "error of the zero model": at this point in the text it is not clear what this means.
L 122 inconsistencies \Pi_\mu vs \Pi_D: does the representation error depend on the distribution \mu ? if so, this is the error in the learnt value function and what is the distribution under which the representation error is bounded?

**Related work**
The related work seems overall well covered.

**Typos**:
L 48: as it differently than in supervised settings,
L107-109: inconsistencies for real numbers \mathcal{R} , \mathbb{R}
L109 : [0..1]
L253: some training methods / a training method
L376: to fails

---

> ### Author Response · Authors · 2022-08-01
> **Response**
>
> Thank you for the review! We appreciate the time and attention it took to review our paper.
>
> To be fully transparent, as far as current deep RL algorithms, we don't fully know to what extent they suffer from this failure mode during regular training. We know that the algorithms themselves are indeed vulnerable to poor off-policy performance, but we don't fully know to what extent this particular mode is the cause of these failures vs. other factors in distribution shift.  This is an interesting and relevant question, and we're also curious to see if this is the case. We've included some discussion on this topic in the new Appendix C.2.
>
> To answer your questions regarding the trade-offs on the practical benefits of regularization and questions on overparameterization, we ran a few more experiments on two-layer networks with a range of parameters covering both under- and over-parameterization regimes. As expected, when under-regularized, having more parameters reduces error. We also show that the behavior around the small-$\eta$ divergence does not change, even when the hidden layer reaches 64 parameters. We included plots of this as Figure 10, and discussion in the new Appendix C.1.
>
> To answer your questions: We don't know of any solution that is guaranteed to stabilize off-policy training. In the longer term, insights from the credit assignment literature may lead to Emphatic algorithms that are provably robust, or other advances (such as the work by Ray Jiang et al. \[1]) may close this gap.
>
> For now, even though regularization can fail in the ways we illustrate, it remains a reasonable method that (usually) offers a fair tradeoff -- as long as we are careful to check that we are not running afoul of the failure modes we explain in the paper. Based on our experiments, some sort of adaptive regularization scheme or simply trying different values of $\eta$ spanning a few orders of magnitude could ameliorate some of the problems we highlight. (We've added this point to the conclusion.)
>
> \[1] Learning expected emphatic traces for deep RL. Jiang, Ray et al. (2020)
>
> (We'll also fix the typos and clarify the unclear points in our writing. Thanks for pointing these out!)

---

> > ### Comment · Reviewer_EBcQ · 2022-08-08
> > **Thank you for your answer!**
> >
> > Thank you for your answers! The authors have addressed my questions and I appreciate the additional experiments the authors provided.

---

### Meta-Review · Area_Chair_au1L · 2022-08-23

**Recommendation:** Accept
**Confidence:** Certain

**Metareview:**

This paper presents a counterexample-driven analysis of regularization in TD learning with function approximation.

Despite the paper's simplicity, the reviewers unanimously though there was a good contribution being made here, and I agree. Highlights include a clarity of presentation and new insights into what is known as the deadly triad. The reviewers generally agreed that these results are relevant to deep RL today, but would have appreciated more forward guidance.

**Award:**

No

---

### Decision · Program_Chairs · 2022-09-14

Accept